# A New Dataset of People Flow in an Industrial Site with UWB and Motion Capture Systems

**DOI:** 10.3390/s20164511

**Published:** 2020-08-12

**Authors:** Mickael Delamare, Fabrice Duval, Remi Boutteau

**Affiliations:** 1Normandie Univercity, UNIROUEN, ESIGELEC, IRSEEM, 76000 Rouen, France; boutteau@esigelec.fr; 2SIAtech SAS, 73 Rue Martainville, 76000 Rouen, France; 3Normandie Univercity, UNIROUEN, CESI, LINEAC, 76000 Rouen, France; fduval@cesi.fr

**Keywords:** indoor localization, Ultra-Wide-Band (UWB) technology, MoCap, range estimation, safety, workers flow, industrial plants, dataset

## Abstract

Improving performance and safety conditions in industrial sites remains a key objective for most companies. Currently, the main goal is to be able to dynamically locate both people and goods on the site. Security and access regulation to restricted areas are often ensured by doors or badge barriers and those have several issues when faced with people being in places they are not supposed to be in or even tools of objects being used incorrectly. In addition to this, a growing use of new devices requires precise information about their location in the environment such as mobile robots or drones. Therefore, it is becoming essential to have the tools to dynamically manage these flows of people and goods. Ultra-wide-band and motion capture solutions will be used to quickly identify people who may be in unauthorized areas or performing tasks which they have been uninstructed to do. In addition to the dynamic tracking of people, this also overcomes some issues associated with moving objects or tools around the production floor. We offer a new set of data that provides precise information on worker movement. This dataset can be used to develop new metrics regarding worker efficiency and safety.

## 1. Introduction

In a factory, or an industrial site, different resources (people, machines, etc.) cooperate and interact to perform various tasks within industrial processes and lead to the realization of different production or support cycles [1]. However, some unintended interactions between these resources may be counterproductive to the objectives of performance, safety and/or security of the company. For example, some items may represent a danger to some operators if they are not equipped with adequate protection (security issues). Other items, which may be of a confidential nature, must not be available to all operators and accessible only to a few of the stakeholders on the site. On the other hand, spatial and temporal monitoring can be used to better control production, storage of products or interventions concerning other operations such as maintenance, hardware or software updates (upgrades), etc. There is a real need to know and control the location of the different items and people in a factory to avoid risks related to people and items in an inappropriate location sometimes (risk of malfunction, risk of leakage of confidential information, risk of security problems, etc.). In practice industrial processes, “organizational methods” are used to schedule the tasks of each operator and plan the presence of certain items (raw materials, finished products, products in stock, tools, spare parts, etc.) at certain locations (line of production, store, etc.). Even if this information is integrated into Enterprise Ressource Planning (ERP) tools [2], no procedures are in place for the presence of people and items within inappropriate locations, thus not excluding the risks mentioned. Besides, these methods are static in the sense that the spatiotemporal location information of objects and people is not updated systematically, quickly, and reliably. The factory of the future, which embodies the 4th Industrial Revolution, is a factory whose purpose is to be flexible, connected, and intelligent [3]. All the components (machines, operators, objects, transport systems, etc.) that make up the company must be able to communicate with the supervision system and between them. Several types of information sources may be used to detect states and locations of people and objects such as sensors, already present in the factory or worn by people, staff schedules, etc. [4]. However, the current technological means are often operated separately by the company’s information systems, not leading to an in-depth, accurate analysis, and a relevant decision on possible risks; this could lead to a security breach. It is therefore necessary to be able to merge all this data from several sources when linking information on the identity of individuals and the nature of objects about their positions. Information on the position will occupy a place around which all other information will be dynamically structured.

This work is constituted of a complete dataset of six workers in a Non-Line-Of-Sight (NLOS) industrial assembly line for three hours, and the creation of an algorithm to smooth worker’s data. We also proposed a classification of area depending on the environment, and an analysis of a worker’s rig. Two modalities are proposed: an ultra-wide-band (UWB) system and a motion capture system called MoCap.

This paper is divided in six parts. We established a survey on existing datasets in NLOS. We presented our experimental set-up. We studied the behavior of our raw synchronized data and results. We proposed a way to improve our data. We proposed our dataset used and interpretation. We presented our further work.

## 2. Survey on Wireless Technologies and Existing Dataset in Nlos

### 2.1. Survey on Wireless Technologies

In the context of industry 4.0, it is necessary to be able to locate the operator in a wide environment and with a faithful accuracy for human machine interaction. We decided to focus our research on wireless technologies instead of cameras because, for example, they need to have a Line-Of-Sight (LOS) and that is impossible in industrial conditions because workshops or buildings are in the majority of cases in NLOS conditions [5,6,7].

The IEEE 802.11 standard (WiFi) is primarily used to provide networking capabilities to different devices in private, public, and commercial environments. Most of the current smart phones, laptops, and other portable user devices are WiFi enabled, which makes WiFi an ideal candidate for indoor localization. However, WiFi is commonly used for communication and not localization, therefore novel and efficient algorithms are required to improve their localization accuracy [8,9]. Industrial sites are not fully covered by WiFi, WiFi requires complex processing algorithms, and is prone to noise. UWB is interesting thanks to its wide bandwidth which makes it more accurate in industrial warehouses.

Radio-frequency systems, such as Zigbee, Bluetooth, Digital Television, Cellular networks, Radar, FM radio, and Phones, based on Digital Enhanced Cordless Technology have an accuracy of about several meters and can cover 10–1000 square meters. However, performance levels and applicability vary widely depending on several factors such as the use of pre-existing reference infrastructure, pervasiveness of devices, signal ranges, and power levels [10,11]. In optimal conditions these systems have an accuracy of about 1 m and can cover a building, UWB systems can provide much better accuracy especially in NLOS situations as UWB is not affected by multipath due to its wide band at 3.5 GHz.

RFID can be very effective. By emitting low-frequency signals (125 kHz), this technology makes it possible to locate indoors with a dm-m level of accuracy and can cover 1–50 m. The accuracy of an RFID system is directly related to the density of tag deployment and reading ranges [12]. Most RFID systems rely on proximity detection of permanently mounted tags to locate a person. This system is expensive to use in a large area due to deployment of tags. RFID systems cannot do trajectory tracking because most of them rely on proximity detection of permanently mounted tags to locate mobile readers [13,14]. UWB will be more interesting because it needs fewer tags and it uses higher bandwidth (above 3.5 GHz), which leads to a better accuracy.

Localization systems based on the propagation of sound waves have an the accuracy of approximately 1 cm and can cover 2–10 square meters through the use of the Time-of-Flight localization technique [15]. Sound, being a mechanical wave, propagates through air and building materials; this means that this technology will be heavily affected by the layout of the facility [16,17]. Sound pollution is also a major concern when working in an industrial environment, meaning that in practice those systems will have a much lower accuracy when used in those environments. Finally, those systems are highly dependent on the sensor placement where UWB is immune to this issue due to its high bandwidth making installation easier [18].

Visible light localization techniques use light sensors to measure the position and direction of the LED emitters. An accuracy of 4 cm has been reported and people can be tracked up to a distance of 5 m [19] and centimeter accuracy in a retail store [20]. The multipath errors drastically reduce the localization accuracy; this technology also requires a Line-of-Sight (LoS) between transmitter and receiver to function properly. In industry, LOS is not guarantee, which is why the UWB system is more interesting, as it uses radio frequency and has a wide bandwidth that can handle NLOS.

Low-power networks are relatively new. They appeared as a need for the Internet of things to develop. Their goal is to achieve a long range low power network so that the sensor’s batteries can last months or years (depending on the duty cycle). There are several protocols in use as of today. The most present in real deployments are Sigfox and LoRa. In particular, LoRa’s nodes have already implemented this geolocation ability. It gives resolutions of about ten meters [10,21]. Long distance between base station and device can sever outdoor-to-indoor signal attenuation due to walls affecting the accuracy of the system of localization. We decided to choose UWB due to its capability to pass through walls, equipment, and any other obstacles thanks to its wide bandwidth.

Magnetic field-based localization has centimeter accuracy and can cover an area of 10 m [22]. Different approaches use the magnetic fingerprint. Information on users’ spatial and temporal occurrences and the magnetic fingerprint with indoor location via Wi-Fi RSS are used to estimate the position of people [23]. The use of magnetic data from several smartphones to generate a magnetic pattern is used for better position detection [24] as well as a multisensor fusion, based on the magnetic field [25]. The use of deep neural networks (DN) to perform magnetic field-based indoor localization using heterogeneous devices [26] allows a more reliable system. An efficient data collection method based on walking and responds to all challenges in the magnetic field as been made for indoor localization [27]. By capturing the unique magnetic signatures of different corridors using a smartphone, they have exploited the magnetic fields present inside as a solution to the location problem. The details of the classification indicate the corridors can be distinguished with a good success rate [28]. The magnetic field can be disturbed by the magnetic field induced by electric motors inside industrial buildings.

We made the decision to focus on UWB technology because it can cover a 50 m area and a precision of approximately 10–30 cm as shown in [10,29]. UWB is also cheaper than other technologies and operate with adequate precision in Non-Line-Of-Sight (NLOS) conditions. It is also much more resilient to multipath when compared to narrow-band as UWB transmits short pulses over a wide bandwidth (3.5–6.5 GHz) [18,22]. UWB also has the unique ability to carry its signals through doors and other obstacles that would reflect signals with a more limited bandwidth and a higher power level [30].

Several different methods are used for location techniques and algorithms in wireless-based localization. Location detection techniques can be divided into three general categories: proximity, triangulation, and scene analysis. There are four methods to calculate position in triangulation categories: techniques based on the measurement of the propagation-time system (Time Of Arrival (TOA), Received Signal Strength Indication (RSSI), and Time Difference Of Arrival (TDOA)) and RSS-based and received signal phase methods are called lateration technique [31]. The AOA estimation technique is also called an angulation technique. Detailed of these techniques can be found in [32].

Dead reckoning is the process of estimating known current position based on last determined position and incrementing that position based on known or estimated speeds over elapsed time. Recent works, such as those in [33,34], present several improvement of this method.

Most indoor localization approaches adopted fingerprint matching as the basic scheme of location determination. The main theme is to collect features of the scene from the surrounding signatures at every location in the areas of interest and then build a fingerprint database. The location of an object is then determined by matching online measurement with the closed location against the database [35].

The proximity detection technique is based on the examination of the location of a target object with respect to a known position or an area. The proximity location technique needs to fix number of detectors at the known positions. When a tracked target is detected by a detector, the position of the target is considered to be in the proximity area marked by the detector [7,36].

Based on the work of Farid et al. [32], Hu junsheng [37], and Khudair et al. [7], we summarize in Table 1 the advantages and disadvantages of the localization method.

### 2.2. Survey on Existing Dataset

Datasets using a motion capture system and UWB exist. Bregar et al. [38] present a UWB position dataset in seven different locations such as offices, bedroom apartments, and a small workshop. Seven location are presented in NLOS; we proposed a dataset of person flows during an assembly phase in a realistic industrial environment.

Cung Lian Sang et al. [39] proposed an experimental dataset for multilabel classification results of a UWB ranging system. We proposed experimental data that reproduce the real localization of workers in an industrial environment.

Kevin Minne et al. [40] proposed data and results related to static measurement of the optimal hardware position on a cyclist for indoor track cycling. This hardware uses ultra-wide-band technology to track cyclists during indoor track cycling. We proposed dynamic indoor localization of workers in an industrial environment with a MoCap as ground truth.

Raza et al. [41] proposed a dataset using UWB, BLE, and a motion capture system covering two scenarios; these being walking and trolley in a room. We proposed two modalities an UWB system in NLOS and a motion capture system in industrial conditions with six different scenarios.

Jorge Pena Queralta et al. [42] present a dataset for UWB-based localization of aerial robots including data from an autonomous flight with an Unmanned Aerial Vehicle (UAV). The ground truth is using an Optitrack motion capture system. We proposed six scenarios of workers flow instead of UAV with same modalities.

Valentín Barral et al. [43] developed a realistic software simulation tool. They have also built a UWB simulator based on real scenarios under three different situations: LOS, NLOS soft, and NLOS hard. They employed machine learning techniques to generate the different models capable of providing simulated UWB ranging. Moreover, the authors of [44] provide a UWB NLOS dataset for NLOS identification and mitigation in the middle of two corridors and a contiguous room; we proposed six dynamics scenarios of workers in an assembly line in NLOS for UWB. We proposed experimental data in real NLOS conditions with a motion capture system as ground truth.

Jiaxin Li et al. [45] proposed the fusion of UWB and inertial data with an Extended Kalman Filter (EKF) to achieve high-accuracy three-dimensional localization. Their dataset, however, contains data from a single flight. We proposed six different workers in industrial environment.

Bernhard Großwindhager et al. [46] proposed a dataset that contains an extensive measurement campaign in two complex indoor environments with one UWB anchor. It contains line-of-sight as well as non-line-of-sight situations. We proposed an indoor industrial situation with six workers and a MoCap system as ground truth.

In relation to these datasets, we offer an NLOS experimentation of six workers in representative industrial conditions, on assembly stations for three hours. By comparing previous datasets, we provide six different scenarios in a realistic environment. We provide real behavior of workers during assembly phase. The environment of workers is surrounded by metals and concrete with obstacles.

## 3. Experimental Set-Up

### 3.1. Follow-Up of People in a Manual Manufacturing Workshop

To obtain data from UWB tags, we placed four anchors in the room used for indoor localization. To be as accurate as possible, we placed the anchors in a rectangular configuration as recommended by the manufacturer, for this we use the MDEK1001 from Decawave’s Real-time locating systems (RTLS), which is shown in Figure 1 as a red rectangle.

We chose one anchor as reference (initialization at x = 0 and y = 0 according to the UWB reference), and we must obtain the position of each anchor according to the initialization anchor as shown in Figure 1. To do so, we use a Two Ways Ranging (TWR) algorithm provided by Decawave. The TWR algorithm from IEEE standard P802.15 [47] estimates the distance between each anchor and the tag as described the following formula,
$$T_{TOA}=\left(T_{2}^{R}-T_{1}^{R}\right)-\left(T_{2}^{T}-T_{1}^{T}\right)-E_{2}-E_{1}$$
where $T_{TOA}$ is the time of flight between reference station and the tag; $T_{2}^{R}$ and $T_{1}^{R}$ are time stamps of the receptor; $T_{2}^{T}$ and $T_{1}^{T}$ are the time stamps of the receiver; and E2 and E1 are the errors affected by the signal power of, respectively, timestamps $T_{1}^{T}$ and $T_{2}^{T}$. If it is assumed that the speed of radio waves through air is the same as the speed of light c, then the distance between the anchors and the tag can be determined by
$$Distance=c\times T_{TOA}$$

The position of the tag can be calculated by TOF measurements from different sensors. The locating engine uses the maximum likelihood between the four anchors to give the position. UWB tags are placed in the pocket of each person.

### 3.2. Industrial Setup

The aim of the process scenarios of the assembly line is to build six tricycles in three hours. Each rig has its own process of assembly. There is one worker for each rig. The stations are fixed as shown in Figure 2b for stations one, two, and three and Figure 2a for stations four, five, and six. During all the processes each person does the same task.

The person at station 1 builds the lower frame of the tricycle. As soon as she finishes, she gives her part to the person in station two. The person at station two receives the lower part of the tricycle and assembles the axle with the lower part. Once she has finished, she gives her part to the person at station four. The person at station three assembles saddle and pedal board and gives it to the person at station four. The person at station four assembles the two parts given by the station, which are the rear, wheels, and axle unit. She will then give it to the person at station six. The person at station five assembles the front wheel of the tricycle and the axle unit. The person will then give it to the person at station six. The person at station six assembles the last two parts of the tricycle.

At the end of the three-hour registration period, six tricycles will be built as seen in Figure 3. Each person will have to restock as the initial stock is for three bicycles. People will have a path deviating from the protocol of the Figure 4, for example, when they go on break. This may mean that the stock was not correctly planned in the first place and the operator will have to replenish it. He/she will therefore take a different route from the initial protocol. People can also go to help other people or take breaks. Thanks to the data of our sensors, we can detect when a person does not respect the assembly process. Actions can also be taken to improve the comfort of the operator.

### 3.3. Motion Capture System

We used a motion capture system from Optitrack similar to that in [48] with millimeter accuracy [49] as a ground truth.

This tool does not directly give the position of people but reflective points. It is by combining them in shapes determined by groups of 4 or 5 (called “solid bodies”) that identification becomes unique and makes it possible to track an object or an individual. The disadvantage of this type of location is the masking of one or more points, making identification unique or impossible. In our case, the metal structures of the manual stations block many lines of sight and therefore it happens regularly that we lose the location. In the database, the result is simply a hole in the data until all reflective points are recovered to ensure the uniqueness of the object or person detected (from 4 points). We have chosen to place reflective dots on people’s heads, which limits data loss.

The table, therefore, consists of a synchronous time column (every 100ms) and the coordinates of the center of each solid body. We have chosen to put the data in the form of an Euler angle for its readability and x, y, and z coordinates.

We had to apply the necessary translations and rotations to move the axes and center as shown in Figure 5. Thus, the Z axis of the experiment becomes the X axis of the database, and the X axis becomes the Y axis. A translation of 5.6 m on the X axis and 5.5 m on the Y axis are finally applied. We do this change because the MoCap system has its proper reference in the workshop and the UWB system also has its proper reference in the workshop. We used several static positions to calibrate the MoCap system in time and positions. At the beginning of the recording all people start at the same position and at the end of an-other same position.

### 3.4. Ultra-Wide-Band System

This tool directly gives the position of the badges worn on the belt or in the pocket of people in the workshop. To exploit the data, it was simply necessary to realign the data on the axes shown in Figure 1, so we reversed the two axes and did a translation of 6.85 m on x and 8.19 m on the y axis. The optimal operating area is given by the red frame in Figure 5. The data remain accessible outside this area, but there is no guarantee of accuracy. Data is not received synchronously. Each badge will provide its position every 100 ms more or less 10 ms if this badge is correctly located. No data will be transmitted if the badge is not correctly located.

### 3.5. Discussion on Raw Data

These two modalities means (MoCap and UWB) each have their means of localization, so it is difficult to guarantee that the axes are perfectly aligned because of the size of the system (working area of more than 50 m^2^). We know that an error in the positioning of the UWB reference badges, or the referencing of these positions directly induces an error in the positioning depending on the position of the badges. The MoCap guarantees positioning accuracy of a few millimeters, but only when it is in use. The very high number of data losses during measurements shows the limits of this system in a dense manufacturing workshop in masking elements. All comparisons are made against the MoCap and only when the location is good. There is no interpolation on the intermediate positions in order not to distort the results. The synchronization of the two means has been validated for an offset of 2292.32 s because the UWB system started before the MoCap system.

### 3.6. Dataset

All sensor readings of a sequence are zipped into two single files named “Filtered_datas_UWB.zip” and “Raw_datas_UWB_Mocap.zip”. The directory structure is shown in Figure 6. We provide raw data from the Motion capture system, which can be compared to the work in [48] as a motion capture system with a millimetric accuracy and UWB system with centimeters accuracy [29,50,51]. Timestamps are stored in timestamps.txt and data format are stored in dataformat.txt. Each line in timestamps.txt is composed of the date, time in hours, minutes, and seconds. Each line in the data folder provide X position and Y position for UWB system and X, Y, and Z position for the motion capture system. A snapshot of the dataset can be seen in Figure 7, with an example of the Rig1Mocap_raw. All zipped file provide six rig position corresponding to the Figure 4. We provide also filtered UWB data which are data filtered by the Savitsy–Golay filter.

The dataset is available at this link: IndoorIndustrialLocalisationDataset: https://github.com/vauchey/IndoorInsdustrialLocalisationDataset/.

## 4. Results and Improved Results

### 4.1. Positions

Figure 5 shows all the measurements over the entire time of the study. The red frame represents approximately the zone of confidence in the UWB location. We can observe many exits from the area by the actors of this experiment. They will not be considered because the MoCap system can only detect the movement of people in the area of the red square seen in Figure 5. Stations 2–4 regularly bypass the area due to an equipment that blocked the space between stations 2 and 5. In these graphs, segments are drawn between the non-validated points, which makes the schema cumbersome but facilitates comparison.

We can see that despite the noise of the UWB’s location, the curves follow each other very well throughout the experiment.

### 4.2. Accuracy

We quickly admitted that it was impossible to have a simple expression of precision. Indeed, several parameters come into play and it is currently tough to know which parameter is predominant over the other. In static phase we can have an accuracy at different localization.

From this observation we have displayed in the form of an image the precision according to the position of the people. We used RMSE computed as follows,
$$RMSE=\sqrt{\frac{1}{N}\sum_{i=1}^{N}{\left(\hat{\theta_{i}}-\theta_{i}\right)}^{2}}$$
where the average difference between the ground truth values from the Mocap system, written $\hat{\theta_{i}}$, and the estimated values from the UWB system expressed by $\theta_{i}$. We compared both modalities UWB and Mocap after time shifting of data. The result is shown in Figure 8.

Points indicating a zero error are those where, there are no data available. For the others, we see a strong relationship between being under the reference tags (at the 4 corners of the red zone in Figure 5) and in the alignment of the metal structures of the substations. The combination of the two makes the strongest errors, such as the coordinate points (8, 3) and (6.5, 9) for example. The maximum error in the free zone is 40 cm, while in the metalized zone we have one meter of error.

### 4.3. Discussion of Raw Values

The impact of travel speed also seems to be very interesting to quantify. The speed is calculated by $V=\frac{d}{t}$, where $D_{n+1}$ is the position at $t_{n+1}$. $D_{n}$ is the position at $t_{n}$ and $t=t_{n+1}-t_{n}$ and corresponds to the timestamp at n and *n* + 1. We only estimate the speed from the Mocap system, because it is our ground truth. The objective of qualifying the UWB for a location of people or objects is achieved. Accuracy of location without understanding the results is close to one meter in the worst cases, and the better understanding of it now can easily improve it. Previous works [29,52,53] show that UWB can be very accurate in LOS industrial conditions however we are in NLOS industrial conditions. Studying the speed during a trip may improve results. The dilution of accuracy caused by the body and the environment (concrete and metal) [54] may also have an impact on the calculation of triangulation.

We calculate the GDOP value as described in [55],
(1)$$GDOP=\frac{RMSE_{loc}}{RMSE_{range}}$$
the result is shown in Figure 9.

One can see a correlation between the travel speed in Figure 10 and the calculation of the Geometric Dilution Of Precision (GDOP) in Figure 9. The anchors are placed at the corner of the black rectangle in Figure 9. We can see that the worst GDOP is in a rectangular plane at the position of the anchors, maximum GDOP is in purple (~5.5), and cyan means that no data were available. The combination of speed and high GDOP gives all the explanations for calculation errors in NLOS. GDOP just gives us an indication of geometry, the calculation takes into account the measurement of noise because we are in NLOS and also a DOP due to the person wearing the tag. However, we can see that the biggest measurement error is in the plane on the side of the rectangle, see Figure 9. This shows that in spite of the measurement noise, the environment and the person, the geometry has an influence on the calculation of the position. The speed also has an impact on the position calculation. Both combined show the majority of the position errors seen in Figure 11. Other position errors that are not mainly due to these two factors are those of the industrial environment. Solutions that could improve the calculation would be to take into account the geometry and therefore the positioning of the tags, which must be placed in the corners of the parts, as high as possible. When there are areas where the person is going to move fast enough, we can add an anchor. Algorithms with knowledge of the environment will be able to correct the errors automatically. Ye et al. [56] proposed a way to place anchors and have good GDOP.

### 4.4. Improved Results

We propose a first method to discard bias calculation in NLOS by using the Savitsy–Golay filter [57] that smooths the trajectory when the speed is high. The Savitsky–Golay filter is defined by the equation
$$\hat{Y_{j}}=\frac{\sum_{i=-m}^{i=m}C_{i}Y_{j+i}}{N}$$
where *Y* is the original position value, $\hat{Y_{j}}$ is the resultant position value, $C_{i}$ is the coefficient for the ith position value of the filter, and *N* remains the number of convoluting integers. *N* is equal to the smoothing window size (2m+1), with *m*∈N. The index *j* is the running index of the original ordinate data table. The smoothing array is constituted of 2m+1 points, where m is the half-width of the smoothing window. The coefficients of a Savitsky–Golay filter ($C_{i}$) can be obtained from Steiner et al. [58]. Two parameters must be determined according to the position as shown in Figure 11. The first parameter is *m*, the half-width of the smoothing window. A larger value of *m* generates a smoother result. We selected five points (m=5) considering that new positions are provided by the UWB at a 10 Hz rate, the combined length of the window will be 11 points (2m+1), each second we will maintain the current corrected trajectory, which is acceptable for a real time application. The second parameter represent an integer (*d*) specifying the degree of the smoothing polynomial. We chose d=1 to consider a linear approximation and discard bias. As shown in Figure 12, when the Savistsy–Golay is not applied shown in Figure 12a. We can see that the trajectory is noisy as in Figure 12b. While the filter is added the trajectory is smoothed. This filter improves the overall accuracy of the UWB seen in Table 2 by one centimeter and discard wrong values, we compare UWB data with the MoCap system when data are trustable. This method can be used in real-time and does not require others sensors such as those in [59,60,61].

## 5. Use and Interpretation

With this dataset, we provide three hours of recording. Six people at the same time on six various kinds of work stations with representative industrial scenarios compared to others existing dataset shown in Table 3.

This database can be used for predictive maintenance or calculating travel time (number of kilometers obtained for each person). We can also do room optimization to improve operator comfort and production efficiency. All the data allow the making of new algorithms using UWB in NLOS and MoCap as ground truth.

In this study, we defined four types of potential UWB error. The first one is the Geometry Dilution Of Precision (GDOP), which is characterized by the geometry of the anchors positioning. When the tag is in a plane zone of the X axis or of the Y axis a larger error will be characterized, as can be seen in Figure 9. We can see that the error is around the anchor positioning rectangle and is around one meter regarding Figure 8 and have a GDOP between 3 and 5. According to Fevzi Aytaç Kaya et al., they classify rating of DOP (including GDOP) between level 1 and 50. Having a GDOP value between 3 and 5 according to [62] is good and excellent when the moving person is inside the black rectangle on Figure 9. This geometry explains the errors close to the X and Y axis planes.

A second cause of position errors is speed. As shown in the Figure 10, the position errors are directly influenced by the user’s speed.

The Dilution of Precision (DOP), the third type of error, causes a less important global error on the position of the tag; it is negligible compared to the others type of errors. Richa Bharadwaj et al. [63] investigated the effect of random placement of base stations on three-dimensional body-centric localization, and Andrew Fort et al. [64] show that the body can have a small impact and the accuracy.

The fourth type of error is the NLOS. It is clearly seen if we compare it to Figure 8, the position errors are well correlated in speed and GDOP. There is an error point that has not been identified by the GDOP and the speed, this error point is where there is metal in NLOS at the coordinate (8,3).

## 6. Conclusions and Further Work

In this article, we propose a new data set for indoor localization in NLOS with six dynamic scenarios in an industrial site during an assembly phase. We also suggest ways of improvement for future data set in order to compare with other research areas. We have shown that geometry and speed to have an influence on the position calculation. We are introducing a new way to filter the UWB position estimation without merging the data with other modalities.

This study aims to provide a better understanding of the position estimation errors on relying on a UWB medium in real conditions of use.

Both modalities can be implemented for indoor localization. The MoCap system shows a better accuracy than UWB but requires a more expensive and complex infrastructure set-up. For use cases requiring less accuracy, or a faster and less expensive set-up, UWB is an excellent choice to increase safety in industrial environments at a low cost.

Our forthcoming work will involve attaching UWB tags on robotic arms in order to set predefined speeds. We also plan on developing an algorithm that would provide a better accuracy in term of position and speed estimation in a dynamic context. The experimental set-up will be similar, i.e., in a workshop with several work stations. Other sensors will be added to compare various types of algorithms in NLOS. We plan on providing an accurate characterization of NLOS effects and DOP values.

## Figures and Tables

**Figure 1 sensors-20-04511-f001:**
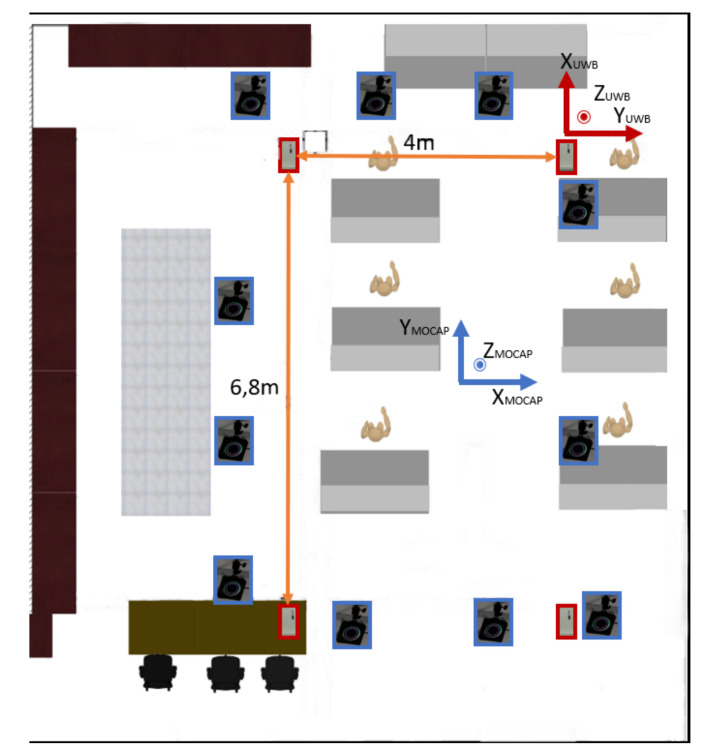
Assembly line set-up. Ultra-wide-band (UWB) anchors are place in a rectangle configuration in red. MoCap cameras are placed around the area in blue.

**Figure 2 sensors-20-04511-f002:**
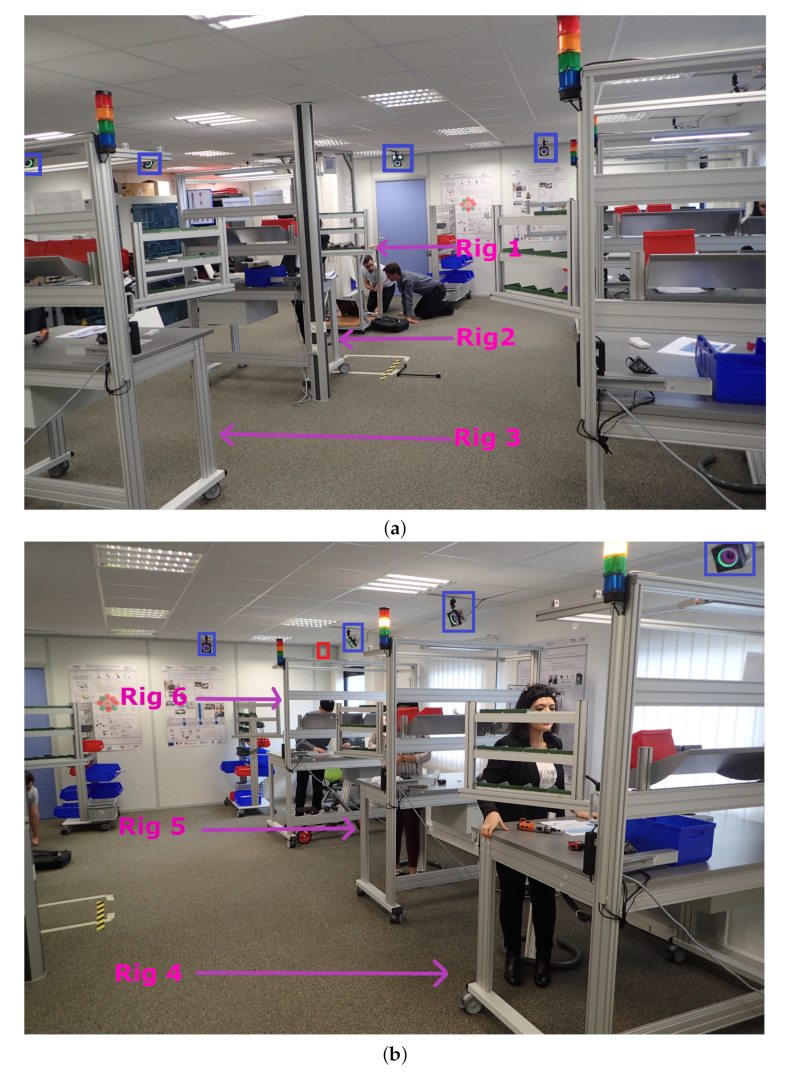
Our UWB set-up in an industrial NLOS assembly line. In the blue squares are the MoCap system and in red square the UWB system when not hidden due to NLOS conditions. (**a**) Industrial set-up in non-line-of-sight (NLOS) with six assembly rigs. view [A]. (**b**) Industrial set-up in NLOS with six assembly rigs. view [B].

**Figure 3 sensors-20-04511-f003:**
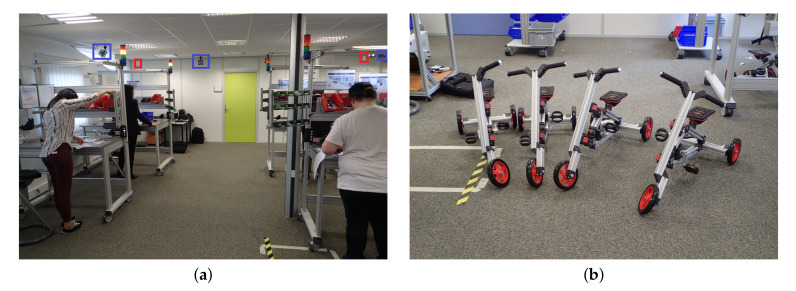
Final assembly of tricycles, and during the process in a NLOS industrial condition made in a workshop. (**a**) Front view of the assembly process. In the red square is the UWB system and in the blue square is the MoCap system. (**b**) Result of the assembly process.

**Figure 4 sensors-20-04511-f004:**
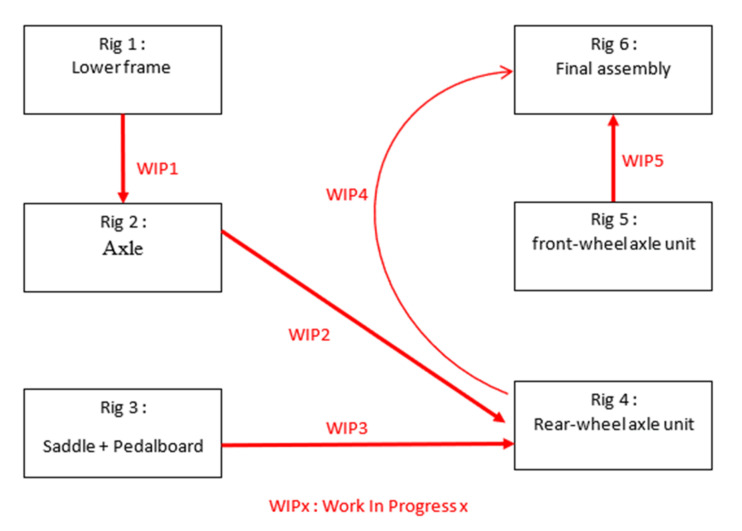
Configuration of the movements scenario of six people corresponding to the six rigs in a NLOS condition made in a workshop.

**Figure 5 sensors-20-04511-f005:**
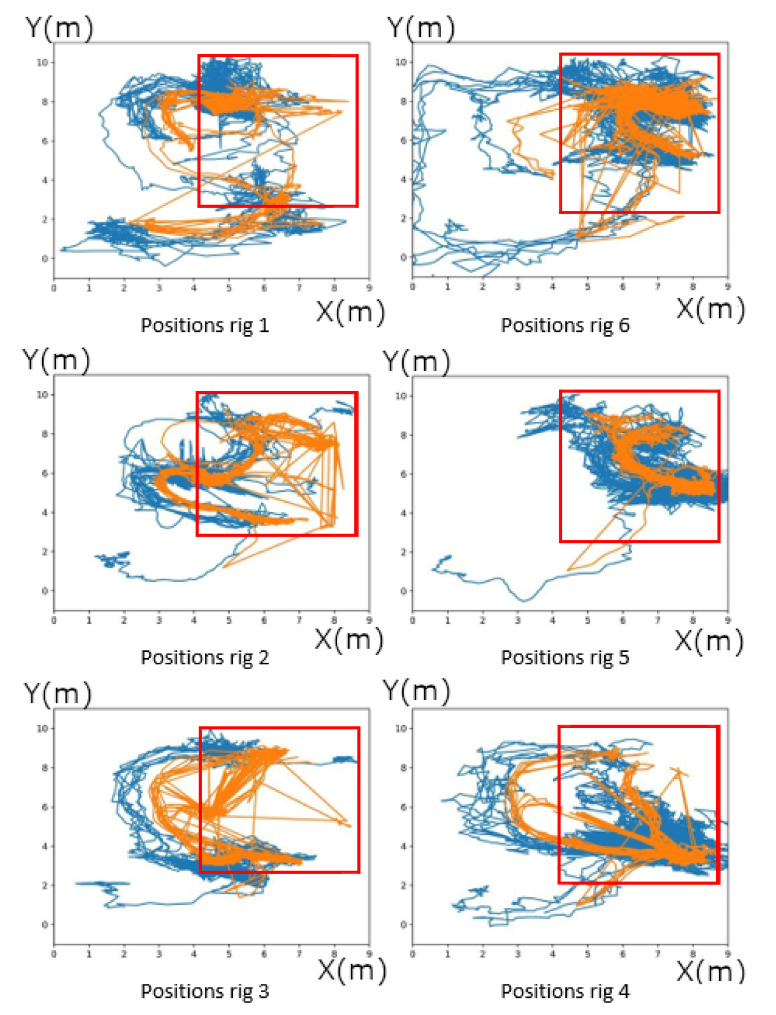
Movement of each person according to their rig in meters made in a workshop. Motion capture system is in orange and UWB system is in blue. Red square is the area of the working assembly.

**Figure 6 sensors-20-04511-f006:**
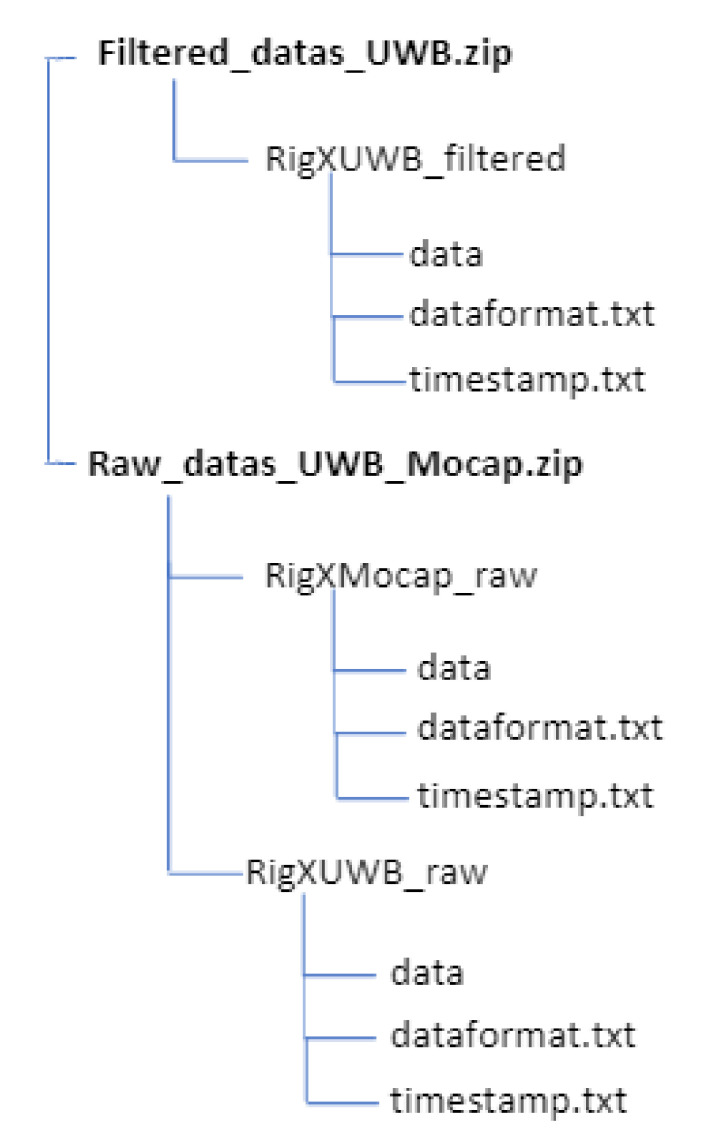
Structure of the provided Zip-Files. X stand for Rig one to Rig six.

**Figure 7 sensors-20-04511-f007:**
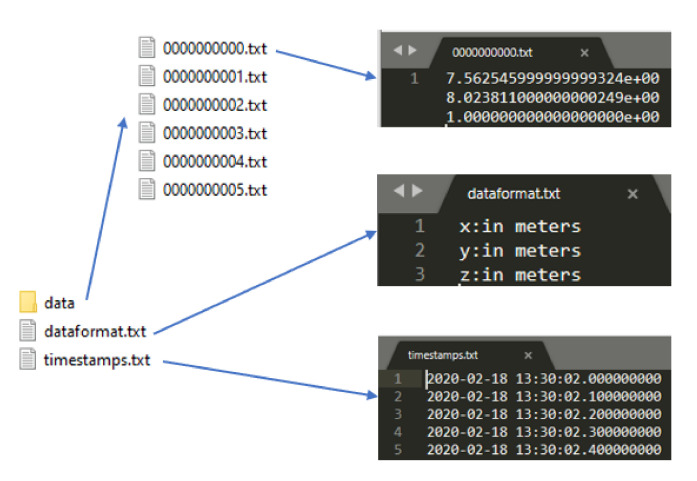
Snapshot of the dataset for Rig1Mocap_raw.

**Figure 8 sensors-20-04511-f008:**
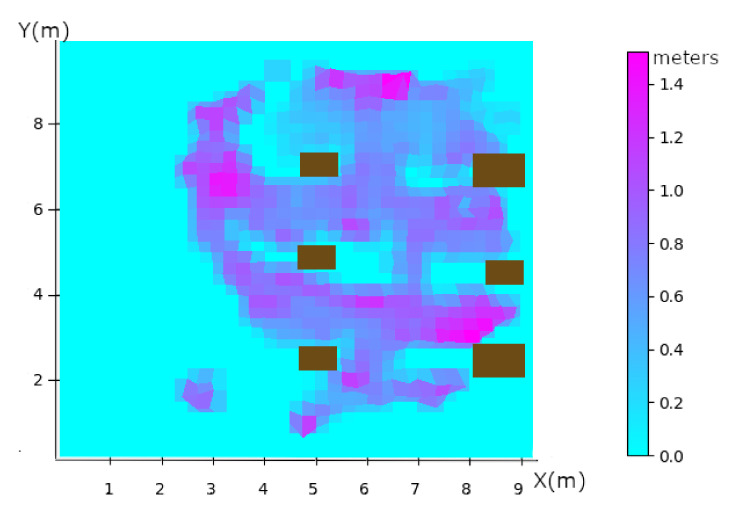
Accuracy in meter as a function of position. Cyan are areas without data (0.0 m). Purple are areas with a maximum error of 1.5 m. Brown rectangle are each rig.

**Figure 9 sensors-20-04511-f009:**
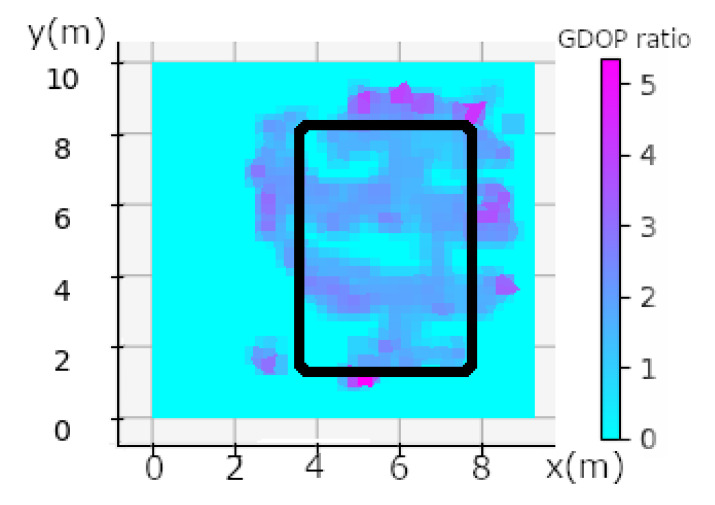
Geometric Dilution Of Precision calculation made in the workshop in the NLOS industrial condition. Black rectangles show the area where UWB anchors are placed, one of them in each corner.

**Figure 10 sensors-20-04511-f010:**
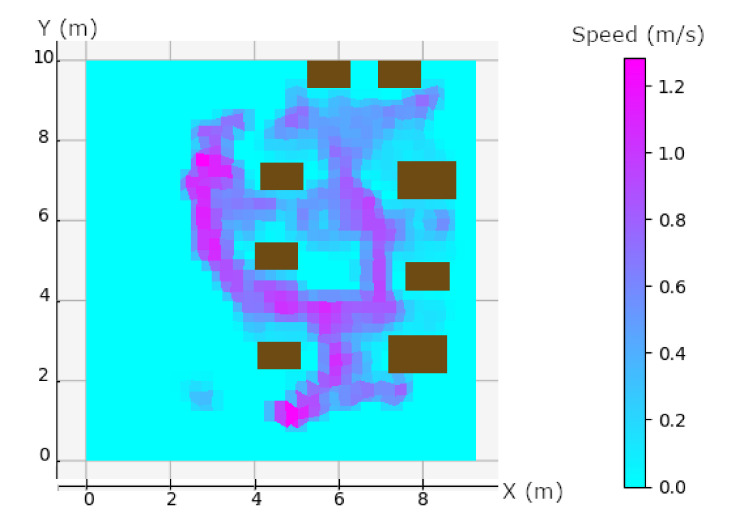
Speed diagram in m/s. Brown rectangles are each rig, the two of the top are the supply rig. Purple values are area with maximum speed and cyan are are with no data.

**Figure 11 sensors-20-04511-f011:**
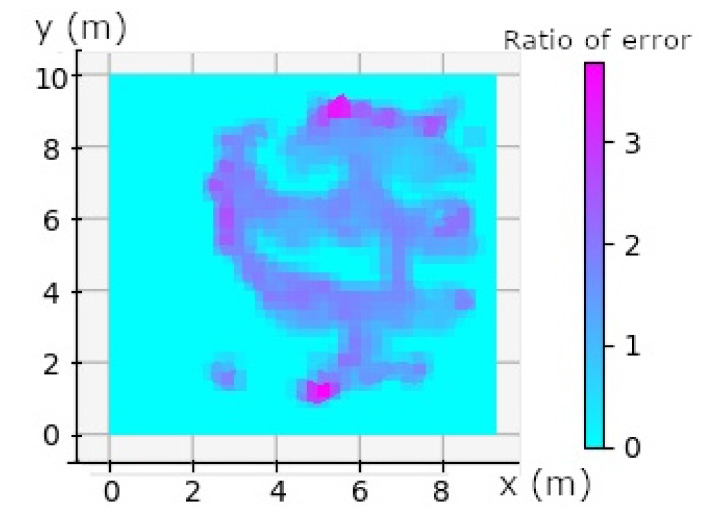
Combined Speed and GDOP ratio. Maximum speed and GDOP are in purple (~4); cyan no data available.

**Figure 12 sensors-20-04511-f012:**
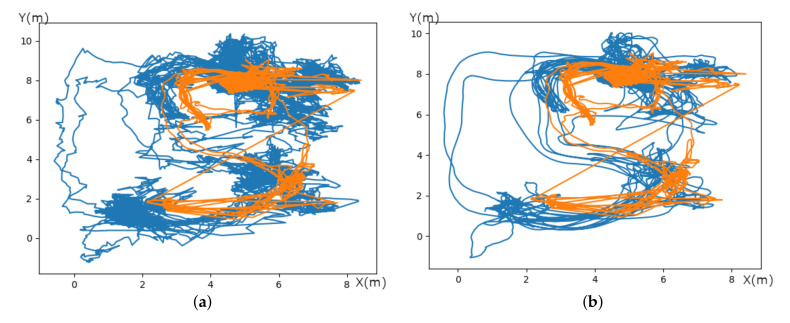
Comparison of the trajectory of the worker from station 1 between UWB filtered and not filtered in blue. In orange the motion capture system. (**a**) UWB system without Sav–Gol filter in blue and motion capture system in orange. (**b**) UWB system with Sav–Gol filter in blue and Motion capture system in orange.

**Table 1 sensors-20-04511-t001:** Indoor localization comparison: techniques and method.

Method	Measurement Type	Advantage in Industrial Site	Disadvantage in Industrial Site	Technology Related to This Method
Proximity	Cell-ID	Accuracy can be improved by adding more antenna. Use for item or access point identifications with high accuracy.	Adding more antenna wil increase the cost. Accuracy depend on the size of the cell. Cannot do trajectory tracking	Wi-Fi, Bluetooth, RFID, Zigbee, Infrared, Visible Light
Direction	Angle of Arrival	Can provide high localization accuracy, does not require any fingerprinting.	Might require directional antennas and complex hardware, requires comparatively complex algorithms and performance deteriorates with increase in distance between the transmitter and receiver. In the NLOS situation for industrial sites, an additional algorithm will have to be used.	Wi-Fi, UWB, Ultrason.
Time	Time Difference of Arrival	Does not require any fingerprinting, does not require clock synchronization among the device and RN	Requires clock synchronization among the RNs, might require time stamps, requires larger bandwidth	Wi-Fi, UWB, Bluetooth, Infrared, Ultrason.
	Time Of Arrival	Provides high localization accuracy, does not require any fingerprinting	Requires time synchronization between the transmitters and receivers, might require time stamps and multiple antennas at the transmitter and receiver. Line of Sight is mandatory for accurate performance.	Infrared, Wi-Fi, Ultrason.
Finger- printing	RSSI	Easy to implement, cost efficient, can be used with a number of technologies	Prone to multipath fading and environmental noise, lower localization accuracy, can require fingerprinting	Wi-Fi, RFID, Bluetooth, Zigbee. Wi-Fi, Bluetooth, RFID, Visible Light, Magnetic Field
Dead Reckoning	Acceleration, Velocity	Can do trajectory tracking with high precision. Not infrastructure -dependent	Inaccuracy of the process is cumulative, so the deviation in the position fix grows with time.	Inertial navigation system

**Table 2 sensors-20-04511-t002:** Comparison with filtered data and raw data with the MoCap system as reference.

	Overall Experiment in Red Square Zone	X-Axis	Y-Axis	2D
Raw UWB data	Mean error	0.21 m	0.12 m	0.16 m
	Range	2.84 m	3.45 m	3.14 m
	Standard deviation	0.46 m	0.38 m	0.42 m
Filtered UWB data	Mean error	0.19 m	0.11 m	0.15 m
	Range	2.74 m	3.44 m	3.09 m
	Standard deviation	0.41 m	0.38 m	0.39 m

**Table 3 sensors-20-04511-t003:** Comparison of existing UWB-based localization and positioning datasets and ours.

Dataset	Distance Est	Modalities	Number of Tag	Anchor Settings	Industrial Scenarii	UWB Node
Cung et al. [39]	AltDS-TWR	UWB	1	4	No	DWM1000
Minne et al. [40]	ToF	UWB	6	8	No	DWM1000
Raza et al. [41]	ToF(TDOA)	UWB+BLE &UWB+MoCap	1	4	No	DWM1001
Queralta et al. [42]	ToF	UWB+MoCap	1–4	multiple	No	DWM1001
Barral et al. [43]	RSS	UWB+IMU +camera	1		No	Pozyx
Li et al. [45]	ToF	IMU+UWB +Mocap(VICON)	1	6	No	TimeDomain
Bernhard et al. [46]	ToF	UWB	1	1	No	DW1000
**Ours**	ToF	UWB+MoCap	6	4	Yes	MDEK1001

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
