# Peer review of "A New Dataset of People Flow in an Industrial Site with UWB and Motion Capture Systems"

_sensors, 2020, doi:10.3390/s20164511_

Round 1

Reviewer 1 Report

The paper presents investigations of the dynamic management of flows of people and goods in an industrial site, which is of interest for the readers. However, there are evident flaws in this paper which are summarized as follows:

  1. The title is too long and it can not clearly reflect the work presented in the paper.
  2. As a research paper, the presented work lacks technical and theoretical depth. The authors should demonstrate how the presented technics or methods can potentially help the industrial system. There should be model to demonstrate how the techniques can fit industrial operations, and why they are useful?
  3. In section 1, there is no single reference.
  4. Very poor organization. For the survey in Section, please put the techniques, there pros and cons in a table, and put the related studies of UWB in a table with taxonomy. The give appropriate discussions.
  5. On reading the conclusion, I can not find anything new and any valuable contributions.

Reviewer 2 Report

Please see the attached file for comments.

Round 2

Reviewer 1 Report

The authors have convered all my concerns. I am happy with the paper in the current form. 

Reviewer 2 Report

All comments have been resolved.